# High-dimensional cortical signals reveal rich bimodal and working memory-like representations among S1 neuron populations
Sofie S. Kristensen, Kaan Kesgin & Henrik Jörntell ✉

Complexity is important for flexibility of natural behavior and for the remarkably efficient learning of the brain. Here we assessed the signal complexity among neuron populations in somatosensory cortex (S1). To maximize our chances of capturing population-level signal complexity, we used highly repeatable resolvable visual, tactile, and visuo-tactile inputs and neuronal unit activity recorded at high temporal resolution. We found the state space of the spontaneous activity to be extremely high-dimensional in S1 populations. Their processing of tactile inputs was profoundly modulated by visual inputs and even fine nuances of visual input patterns were separated. Moreover, the dynamic activity states of the S1 neuron population signaled the preceding specific input long after the stimulation had terminated, i.e., resident information that could be a substrate for a working memory. Hence, the recorded high-dimensional representations carried rich multimodal and internal working memory-like signals supporting high complexity in cortical circuitry operation.

With the increasing use of multi-neuron recording techniques, neuroscience has seen the emergence of new approaches to explain the behavior of neuron populations rather than of individual neurons one-by-one, with the aim to provide better insights into the computational architecture of the neuronal circuitry. But the fact that there are multiple different approaches to describe neuron population dynamics and to correlate neural activity with real-world parameters[1] illustrates that the actual computational architecture of cortical information processing remains an open issue. Therefore, there is a need to further identify features of the physiological infrastructure that underlies the computations in the brain circuitry. One aspect that has been overlooked in the literature in relation to such physiological infrastructure is observations indicating that natural behavior intrinsically has a high complexity, i.e., a meaningful structural richness. In neural systems, complexity can have major advantages from a behavioral point of view as it can confer flexibility and rapid learning. It has been observed that even for superficially monotonous behaviors such as locomotion, healthy individuals exhibit high complexity, across multiple time scales, whereas aged subjects and neurological patients display impaired behavioral complexity[2–5]. Kinematic variability has also been identified at a more detailed level[6]. Through manipulating the sensory feedback in speech, it has been shown that the brain actively generates variability[7]. Also, when the brain actively induces motor variability, it speeds up learning[8].

Considering the pervasiveness of complexity in behavior, it would seem likely that complexity is also a pervasive feature of brain circuitry operation. In general, a complex system is composed of many components, which can interact with each other and thereby create dependencies and other types of relationships that cause distinct system-level properties to arise, one of them being that the system output has high complexity. A complex system can be described as a network where the nodes represent the components and the links between them represent their interactions. In fact, the brain fulfills many of the definitions of a complex system, where the neurons correspond to the nodes and the synaptic connections between them create dependencies. If the individual neurons have at least partially unique activity properties from the other neurons, the central features of a complex system would be fulfilled. But if the neurons are partially unique, that would also mean that the activity distribution patterns across the neuron population would be high-dimensional and thus it would be of great interest to understand that dimensionality better. In contrast, in the current multi-neuron recording era, there is instead an abundance of approaches to perform dimensionality reduction on neuron population data[1,9]. But that approach may result in losing the underlying complexity of the brain circuitry signals. There are studies emphasizing a high-dimensionality of the neuron population codes. However, these studies considered cortical representations purely from the point of view of modality-specific inputs,

Department of Experimental Medical Science, Neural Basis of Sensorimotor Control, Lund University, Lund, Sweden. ✉e-mail: Henrik.jorntell@med.lu.se

i.e., assuming that the activity in S1 neurons is solely a consequence of tactile sensory inputs[10] and that the activity in V1 neurons is solely a consequence of visual sensory inputs[11,12]. Recent studies suggest a strong involvement of cortical internal activity, broadcast across essentially the entire thalamo-cortical system, including the hippocampus[13–19]. Because of its wide base of origin, such internal broadcasting could be expected to result in high complexity in the individual neuron signals, or a high dimensionality in the population-level signals. As a consequence, substantial parts of the neuronal signals in S1, for example, could be unrelated to tactile input but instead reflect intrinsic representation signals, which may be more difficult to interpret but could for example potentially provide for prediction-based perception such as Bayesian inference[16,17,20].

A general challenge for analyzing complexity in the brain is that one needs full control of the sensory input that goes into the system, as the sensory input will impact the precise motor output. But in a system with high complexity, where motor outputs are intrinsically highly variable even for a seemingly repetitive task, this is a potentially insurmountable problem due to the lack of control of the resulting sensory inputs on a trial-by trial basis. Here we used tactile and visual spatiotemporal sensory activation patterns with highly reliable repeatability (see *Methods*) to ensure that any signal variability was due to factors that were intrinsic to the brain circuitry, thus focusing the analysis of the complexity to the brain circuitry. A different challenge is related to the resolution of the neural activity measurement. Because neurons operate as electrical entities, intracellular recording of the membrane potential is a highly resolvable recording method. But it is very difficult to reliably make more than one or two such neuron recordings in parallel in vivo. The complexity in a neural system can be expected to be at least as dependent on the activity distributions across the neuron population as it is on the complexity of the neuron signal. Therefore, multi-unit recordings offer an advantage. Many recent studies have employed imaging of intracellular calcium activity and in this fashion been able to record from up to in the order of 10,000 neurons[11,12,21–24]. One caveat with that approach is that the activity of each individual neuron is characterized at a lower temporal resolution than that provided by electrical measurements of the neuron spiking activity. Since the estimate of complexity depends on the richness of the signal across multiple time scales, here we opted for neuronal spike recordings with the Neuropixels multielectrode array.

If the brain circuitry is considered a system with high complexity, then also its spontaneous activity would be characterized by high complexity, i.e., by a high-dimensional state space[25]. This means that the effect of any given sensory input on the activity of the neuron population will depend on which part of its state space the system is in when the input occurs[16]. Here, we defined the high-dimensional state space of the spontaneous activity of the neuron population and considered the sensory inputs as potentially perturbing factors. The main question asked was whether the perturbation was sufficient to evoke detectable changes relative to that state space, and whether different inputs caused different such changes. With this approach, we found that the S1 neuron population responses to given tactile input patterns changed when they were combined with visual inputs and that the S1 neuron population could even separate fine nuances of visual input patterns. More surprisingly, we found input-specific state changes to tactile input that outlasted the sensory inputs, compatible with a type of working memory.

## Results

We analyzed data from five experiments, each with simultaneous recordings of 36–70 isolated neurons in S1 cortex. In all of the experiments, we analyzed the dimensionality of the spontaneous activity. In 2 of these experiments, we studied tactile input patterns ($N = 4$ patterns) and visual input patterns ($N = 4$ patterns) separately, in 2 experiments we studied tactile input patterns ($N = 4$ patterns) as well as visuo-tactile input patterns ($N = 4$ patterns), and in one experiment we studied both stimulation sets. Hence, each type of sensory input experiment was repeated at least 3 times in total.

## Spontaneous versus tactile and visual activity distributions of the S1 neuron population

We first quantified whether the activity covariance patterns of the neuron population altered relative to the spontaneous activity when there was a sensory stimulation present (tactile or visual). To this end, we applied principal component analysis (PCA) to extract the dominant covariance patterns in the spontaneous data (defined as the activity that occurred in the time windows of 300 ms before the onset of stimulation, the prestimulus time window). Then we applied the sensory stimulation patterns, for example tactile stimulation patterns, and analyzed the data in the 300 ms long poststimulus onset time window (individual stimulation patterns lasted for 200–350 ms) relative to the spontaneous data. Figure 1A shows the average responses for a subset of 25 neurons across all four tactile stimulation patterns (shown in two different formats) as well as the PC scores obtained for the full neuron population recorded in that experiment, across all time points. Note that since the PCA focuses on unique activity distribution patterns across all of the neurons, there is no simple relationship between the evoked responses and the scores of the individual PCs. For example, if all the neurons respond in parallel that is not a new covariance pattern but if a special sub-population of neurons respond differently from the other, that is potentially a unique covariance pattern.

Figure 1B illustrates that the activity distributions within the neuron population overall did change relative to spontaneous activity already for the first 3 PCs, both for the tactile input patterns and, more surprisingly, for visual input patterns, although the latter effect was more subtle. To systematically quantify to which extent evoked data points tended to cluster with other evoked data points we used kNN decoding analysis, which is based on Euclidean distance calculations between the data points (Supplementary Fig. 1). Figure 1C illustrates kNN decoding results when the number of PCs included in the decoding analysis increased from 3 to the number of PCs required to explain 50% and 95% of the variance, respectively. We also performed a fundamentally different analysis of the activity distribution patterns using convolutional neural networks (CNNs) as a control. Also the CNN analysis indicated that the evoked neural activity distribution patterns were different from the spontaneous distribution patterns (Fig. 1C), although at a lower level than kNN results for the full set of PCs. It should be noted that we did not spend any effort optimizing the CNN analysis, we were merely interested in demonstrating that this fundamentally different type of analysis could confirm the results we obtained using PCA. In combination, both these approaches confirmed that analyzing the activity distribution patterns across the neuron population provides a sensitive tool for detecting nuanced systematic changes evoked by conditions that could be expected to have subtle but systematic effects on the neuron population. The PCA further showed that the tool became more sensitive when more dimensions of the data were considered.

For the remainder of the decoding analysis in this paper, we used the full set of PCs required to explain 95% of the variance of the spontaneous data, which in the illustrated experiment (Fig. 1A–D) was 50 PCs and in the other 4 experiments 27, 55, 42, and 28 PCs. This corresponded to 68%, 81%, 79%, 62%, and 78% of the number of PCs needed to explain 100% of the variance. Furthermore, this indicates that the spontaneous activity was extremely high-dimensional relative to the number of neurons recorded in each respective experiments. The accuracy for separating spontaneous activity from evoked tactile and visual activity in the illustrated experiment was 93% for tactile inputs and 81% for visual inputs (Fig. 1D; blue line indicates the accuracy of the CNN decoding in this case). When the data point labels (spontaneous vs evoked) were shuffled the decoding accuracy collapsed to a relatively narrow normal distribution around the theoretical chance level (50%) (Fig. 1D, red curves). Hence, the evoked activity distributions were significantly different from the spontaneous activity distributions.

We next analyzed how different PCs could contribute to the decoding accuracy (Fig. 1E). As expected, the higher the number of PCs included in the analysis, the higher the F1 score. What is noteworthy here is that a high number of PCs were required to reach F1 scores above 90–95% (for the

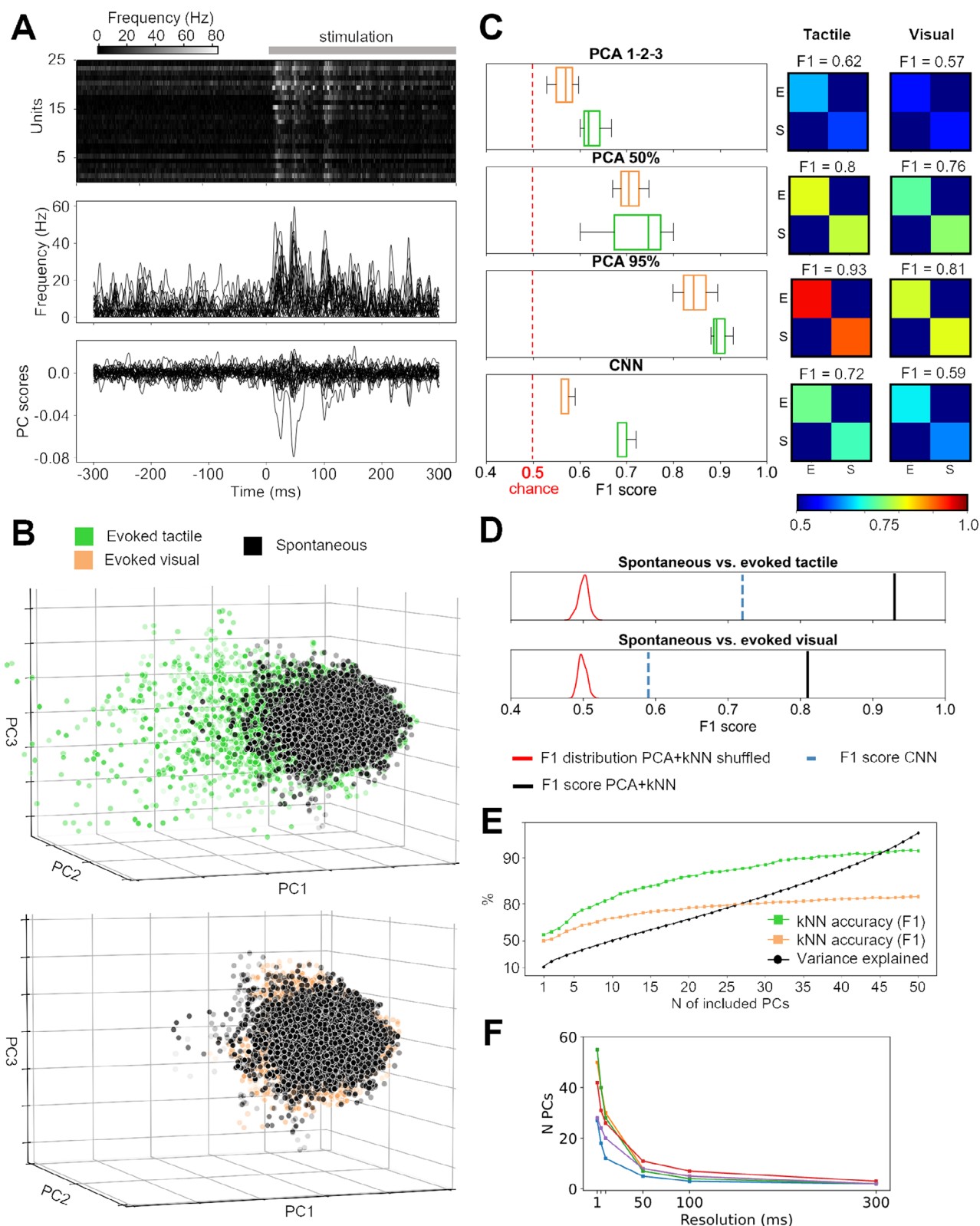

tactile input). This indicates that the information in the recorded neuron population, which was after all a highly limited set of neurons with maximum 70 neurons per experiment, is high-dimensional, i.e., the number of activity distributions that the population can display is extremely high relative to its size. This is a powerful asset in a network, as it means that it has high representational capacity. We also directly tested to what extent the

temporal resolution of the neural recording data impacted these results. Figure 1F illustrates that the dimensionality of the neural data degraded exponentially with lower temporal resolution. The number of PCs needed to explain 95% of the variance as a function of the temporal resolution of the neural recording data is also reported as numbers, alongside the number of neurons, across all experiments in Supplementary Table 1.

**Fig. 1 | Principal component analysis (PCA) showed that both tactile and visual inputs changed the activity distribution patterns of the S1 neuron population.** **A** The responses to tactile inputs for a subpopulation of the recorded neurons from one experiment. Top, normalized averaged spike responses of individual neurons across all tactile input patterns. Middle, superimposed average spike responses of the individual neurons (middle) averaged across all tactile input patterns shown with the spike frequency on the y-axis. Bottom, the PC scores of the full neuron population of the experiment ($N = 61$ neurons) for the first 25 principal components (PCs). **B** Activity distribution patterns as quantified by the PC scores of the first 3 PCs for the S1 neuron population activity during tactile (top) and visual (bottom) evoked responses, compared to the spontaneous activity. Note the same viewing angle in the two plots and that the distribution of the spontaneous data points therefore is identical between the plots. In both plots, the depth location of a data point is reflected by its level of transparency. **C** Decoding performance for visual (orange) and tactile (green) evoked responses using PCs#1-3 (top), PCs explaining 50% of the variance, PCs explaining 95% of the variance, as well as the non-PC based, convolutional neural network (CNN) analysis, across all experiments. Insets to the right are the corresponding confusion matrices from the experiment illustrated in (**A**–**D**). **D** The decoding performances, quantified as the F1 score for PCs explaining 95% of the variance (black line), compared to the results for the same data across 50 different label shufflings (red) as well as the decoding obtained using CNN (blue dashed line). **E** The kNN decoding accuracy (F1 scores) as a function of the number of PCs included in the analysis for visual (orange) and tactile (green) inputs. The black line indicates the cumulative variance explained for each PC. **F** Number of PCs needed to explain 95% of the variance as a function of the temporal resolution of the neural data across the 5 experiments.

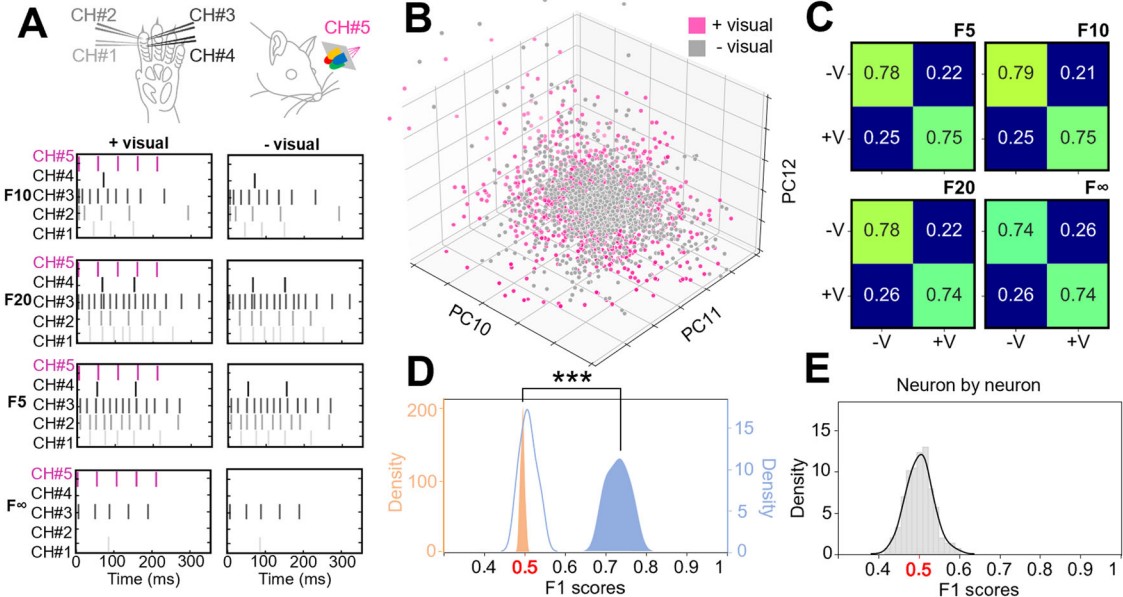

**Fig. 2 | S1 neuron population responses to given tactile input patterns changed when they were combined with visual inputs.** **A** We used a set of 4 fixed tactile input patterns (the labels of which are indicated to the left of each respective pattern, not to be confused with the F1 score) applied to digit 2, which were combined with simultaneous visual stimulation in half of the trials (+visual versus –visual). **B** Neuron population responses illustrated for three example higher-order PCs. **C** Decoding accuracy (F1 scores) for the separation of the two conditions (with or without simultaneous visual input, +V and −V) for the four individual tactile input patterns indicated by their respective labels in the top right corner. The chance level was 0.5, or 50% (as the decoding task was binary, combined or not combined with visual input). Note that the F1 score is based on all PCs required to explain 95% of the variance in this and all other figures below illustrating F1 score results. **D** Across the three experiments and the four tactile input patterns ($N = 12$), F1 scores for the separation of the two conditions (+V vs −V) for non-shuffled data (blue) and shuffled control (orange) (asterisks indicate their distributions to be significantly different at $p < 0.001$). The blue curve without fill represents the non-shuffled data when it was down-sampled to 300 ms resolution. **E** Decoding performance for the same four tactile patterns +V/−V for the 62 S1 neurons when the responses of each neuron were considered separately (as opposed to taking the information of the activity distribution across the neuron population into account).

## Separation of tactile and visuo-tactile input patterns by S1 neurons

In the next experiment, we instead tested if visual stimuli presented simultaneously with a tactile stimulus could alter the responses of the S1 neuron population to the given tactile input pattern (Fig. 2A). The distribution of the data points for some randomly selected higher-order PCs (note that Fig. 1E shows that no specific high order PC contributed more than others), PC#10-12 (Fig. 2B), suggested that the distributions were indeed dissimilar and illustrates that higher order PCs contributed to the separation of the input. Quantitative analysis (kNN) across all the PCs showed that the neuron population could separate tactile input patterns from their respective visuo-tactile input patterns (Fig. 2C). A two-sided paired sample t-test was conducted to determine the difference between the F1 scores and the shuffled F1 scores across all four patterns in all three experiments ($N = 12$). The results indicated that the distribution of F1 scores (mean(M) = 0.73; standard deviation (SD) = 0.027) was significantly higher than the distribution of the shuffled F1 scores (M = 0.5;

SD = 0.002); [t(11) = 29, $p < 0.001$, $d = 10.9$] (Fig. 2D blue curve, filled). However, when the down-sampled data (resolution of 300 ms instead of 1 ms) was used, this separation vanished (Fig. 2D blue curve, no fill). In contrast, individual S1 neurons could not separate these two conditions (Fig. 2E), even though comparing the peristimulus time histograms (PSTHs) of individual neurons was weakly suggestive of such input-condition-dependent differences (Supplementary Fig. 2). In other words, the distribution patterns of the S1 neuron activity contained much more information than individual neurons could provide as long as the higher dimensions of the neuron population activity distribution were also considered.

## Separation of visual input patterns by the S1 neuron populations

We next tested if the S1 neuron population could separate visual input patterns even if there were minute differences between them. For this purpose, we used a set of four spatiotemporal color patterns (Fig. 3A), which were equal length, equal amount of input per color, and only differed with

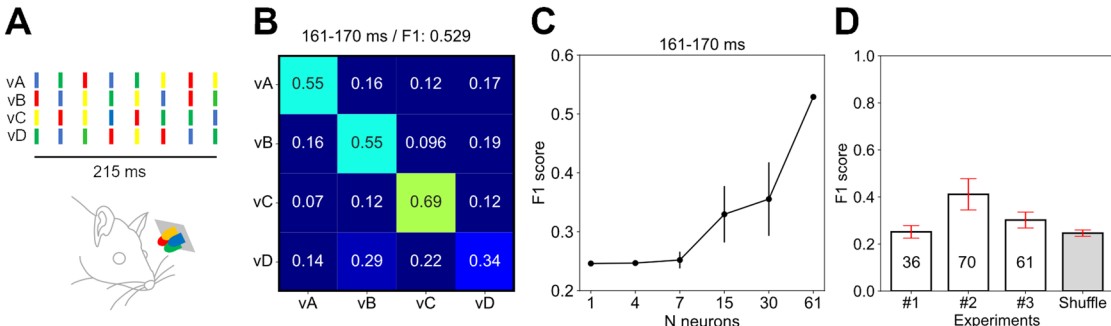

**Fig. 3 | The activity distribution patterns of S1 neuron populations separated fine nuances of visual input patterns. A** The four visual patterns used (vA-vD). The patterns had equal lengths, equal pulse lengths and equal interpulse intervals, but the color series differed. **B** An example confusion matrix of the visual input pattern decoding performance quantified for a population of S1 neurons in one time window, with the time window and the F1 score indicated on top. **C** The decoding accuracy as a function of the number of neurons included in the analyzed neuron population. Each test was made using N random samples from the available neuron population and the mean ± 2 SDs is shown. **D** The average F1 score for the S1 neuron populations across all time windows, shown for each of the three experiments. The standard deviations are shown in red. The chance level decoding was obtained by shuffling the visual input pattern labels and ended up at the theoretical chance level of 0.25 (25% for four stimulation patterns).

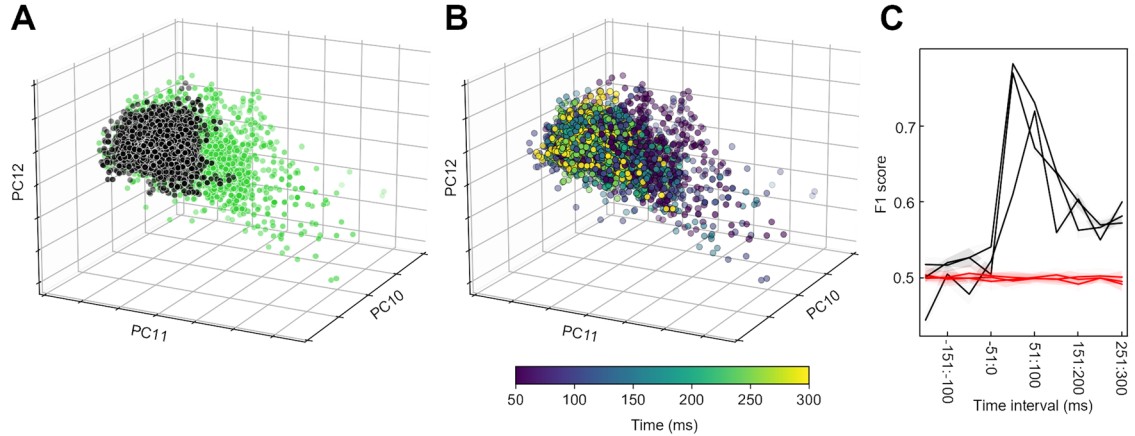

**Fig. 4 | The cortical separation of tactile versus spontaneous data depended on the time window considered. A** Spontaneous versus tactile data for a few of the higher order PCs (#10–12), which is the same data as in Fig. 1B, although Fig. 1B instead illustrates the three lower order PCs. **B** Same viewing angle as in (**A**), but with spontaneous data removed and the timing of the data point versus the stimulus onset being color coded (key at the bottom). **C** Across the three experiments, the average kNN separation (F1 score) and its 95% confidence interval (CI) per each 50 ms time window from 200 ms prestimulus onset to 300 ms poststimulus onset. The plot shows both the actual data (gray) and their shuffled controls (red), for each respective experiment.

regards to the order in which the individual colors were presented. The rationale for this approach is that because the cones and rods have different wavelength tunings[26,27], inputs composed of different colors are guaranteed to result in reproducibly differential patterns of retina photoreceptor activations at the population level, which cannot be guaranteed for non-colored inputs unless high-resolution corrections for shifts in lens focal plane and gaze direction are implemented. This hence constitutes the spatial element of the visual spatiotemporal input patterns. We have previously used similar patterns for analyzing the responses of V1 neurons and found that each V1 neuron encoded these spatiotemporal visual input patterns in highly unique ways[28]. Here, more surprisingly, we found that each visual input pattern evoked at least partly unique responses across the population of S1 neurons (Fig. 3B). When subdividing the 300 ms responses into 10 ms time windows, we could compare the specific temporal variations in the responses with separate kNN analyses for each of the resulting 30 time windows (results from one example is shown in Fig. 3B). We found that the decoding depended on the number of neurons included in the analysis, where a single or a few S1 neurons as a rule could not separate the visual input patterns (Fig. 3C), unlike the V1 neurons[28]. Across the 30 time windows, we found that two out of three experiments displayed a solid separation of the visual input patterns, whereas the experiment where there were the fewest number of neurons failed to show such separation (Fig. 3D). Hence, the decoding

clearly depended on the number of neurons included in the analysis, and these results would not have been detected without the population level analysis.

## The cortical separation of tactile input patterns depended on time after stimulus onset

We also tested if the activity distribution across the S1 neuron population could be used to tell how much time had elapsed after the stimulus onset. This appeared to be the case for the tactile input patterns versus the spontaneous activity (Fig. 4). This prompted a more systematic test of the separation of the four tactile input patterns across each 10 ms time step (Fig. 5). Perhaps unsurprisingly, the separability between the four tactile input patterns gradually rose and then decayed during the approximately 300 ms long time window where the stimulation patterns played out (Fig. 5A, C), which was also true when comparing the spontaneous data with the combined tactile input data (Fig. 4B). What was surprising, however, was that the S1 neuron population continued to separate the tactile input patterns long after termination of the sensory input (Fig. 5C). This was confirmed by comparing the distribution of F1 scores in the 200 ms prestimulus time window to distributions of F1 scores in five different 200 ms poststimulus time windows. The results indicated that the F1 scores in the 200 ms prestimulus time window (M = 0.3027, SD = 0.0347) were

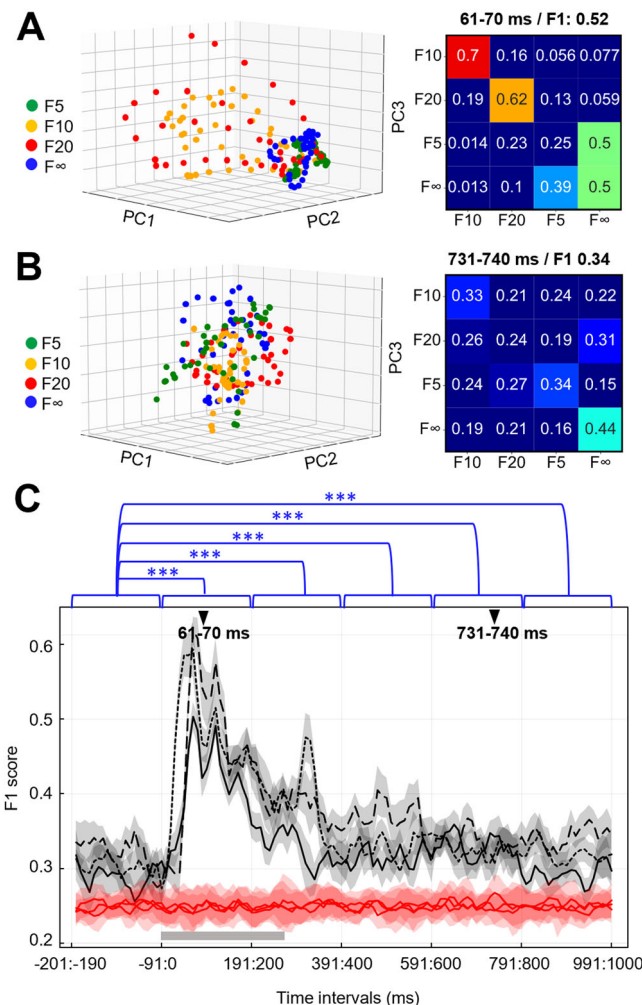

## Discussion

Here we showed that the activity distributions across a limited local S1 neuron population changed relative to the spontaneous state space for both tactile and visual inputs. We also showed that the S1 neuron population responses to given tactile input patterns substantially changed when they were combined with visual inputs, which translate to that the visual input could impact the S1 processing of haptic inputs as a form of anticipatory regulation. The S1 neuron population could even separate different visual input patterns. Aspects of these results were lost when we considered fewer dimensions of the data or fewer neurons, which hence indicates that the representations within the neuron population were high-dimensional. We also showed that the precision of the cortical separation of evoked tactile versus spontaneous data depended on the time window considered and that the S1 neuron population retained input-specific information long after the termination of the stimulation. In fact, after a tactile input pattern the S1 neuron population never returned to the baseline defined by the spontaneous activity, within the 1000 ms time window tested. Our findings imply, relative to many recent studies addressing neuron population coding, that cortical information processing is extremely high-dimensional and incorporates diverse modalities as well as long-lasting signals, reminiscent of a type of working memory, related to internal processing elicited by previous sensory inputs.

In general, the analysis of the activity distribution patterns assumes that the most relevant signals of the neural system reside in the time-varying activity dependencies between the neurons, rather than the independent responses of any individual neuron. In this setting, the resolution of the analysis matters when it comes to achieving estimates of the complexity and dimensionality of the neuron population signals, and therefore for constraining the description of possible computational architectures. Although many recent studies addressing population representations with multi-neuron recordings also analyzed activity distributions patterns, the use of recording methods of lower temporal resolution (i.e., calcium imaging, or binning spike counts into time windows of 100's of ms)[11,12,21–24] would be at risk of averaging out many of the activity dependencies/signal covariance patterns that occur at faster time scales. Our approach was instead to use electrophysiological recordings with high temporal resolution in order to capture also the finer temporal nuances in the covariance patterns in the neuron population. Also, we used stimuli designed to activate the same spatiotemporal set of sensors across multiple repetitions, which allowed a higher precision analysis because any systematic response variation would now solely be due to factors intrinsic to the processing network rather than to potential differences in sensor activation (see Methods). Hence, these approaches put us in an advantageous situation when it came to exploring the intrinsic complexity of the cortical neuron population signals.

In previous neuron population studies, different types of dimensionality-reduction methods were applied to the activity to achieve low-dimensional embeddings, in some cases also to correlate the activity state with some selective task-related parameter. A general problem of behavioral correlation-based approaches is that correlations could well be found without necessarily providing any information or any clear idea of how the problem is structured internally in the brain networks (i.e., its computational architecture). In a highly complex dynamical system such as the brain, if it has a sufficient richness of signaling, the chances of finding at least some correlation with almost any external variable can be high – that is the consequence of complexity. There is also the potential concern that since the thalamocortical system is a positive feedback loop, it may be responsible for generating a dominant part of the population-level covariance, also in the awake state[29]. Low-dimensional embeddings of such data could be at risk of representing the thalamocortical loop activity to a significant extent. The finer nuanced activity representing rich internal processing may need to occur in high dimensions to be useful, i.e., to confer fast and rich flexibility in a complex real-world environment but may be ignored in low-dimensional embeddings (see below). In fact, in our experiments, using anesthesia, the thalamocortical loop activity would be expected to be more synchronized (i.e., this is the reason that slow waves are observed in the EEG under

**Fig. 5 | The S1 neuron population retained information about the specific tactile pattern long after it had terminated. A** Activity distribution patterns across the neuron population (1 expt.) for four different tactile input patterns (color coded) across a 10 ms time window (61–70 ms), exemplified in a lower order PC space (PCs #1–3). Note that for each stimulation pattern there are 40 data points for each 10 ms time window (see methods). Right, the kNN confusion matrix for the same time window. **B** Similar plot as in A for a later time window. **C** For each experiment, the average F1 score and 95% CI for each time window ranging from −200 ms before until 1000 ms after the stimulus onset. Below, in red, the corresponding average F1 score and 95% CI for each time window when the labels of the stimulation patterns were shuffled. The duration of the tactile patterns is indicated by a gray bar, arrowheads indicate the time points for the data illustrated in (**A, B**). The results of t-tests comparing the F1 scores of the 200 ms prestimulus time window with those of the different 200 ms time windows are shown on top. Data from all experiments were combined in each comparison, the asterisks indicate the significance level $p < 0.01$.

significantly lower than the F1 scores in the poststimulus time window 1–200 (M = 0.4546, SD = 0.0941) [t(2999) = −87.65, $p < 0.001$, d = 2.1], the poststimulus time window 201–400 (M = 0.3627, SD = 0.0543) [t(2999) = −59.87, $p < 0.001$, d = 1.3], the poststimulus time window 401–600 (M = 0.3372, SD = 0.0409) [t(2999) = −40.82, $p < 0.001$, d = 0.9], the poststimulus time window 601–800 (M = 0.3289, SD = 0.0283) [t(2999) = −31.03, $p < 0.001$, d = 0.8], and the poststimulus time window 801–1000 (M = 0.3181, SD = 0.0343) [t(2999) = −19.92, $p < 0.001$, d = 0.4]. This indicates that the cortical network retained information about the exact input pattern for a long time after the input had stopped, indicative of an input-specific state memory resident in the activity of the recorded population of neurons.

anesthesia as well as deep sleep), which in turn would make low-dimensional components in the population level activity more prominent (as also found to be the case when sleep and awake states are compared[30]). Despite this factor, we observed the population activity to be extremely high-dimensional – and the dimensionality would hence be expected to be even more pronounced in the awake state, given that the measurements allow its detection. And, as illustrated in Fig. 1F, the functionality-relevant dimensionality of the network also depends on the actual temporal resolution of the information representation within the cortical networks, but that issue would require a very different investigation to be settled.

The spontaneous activity was shown to vary substantially, forming a diversified, high-dimensional state space (Fig. 1). This can be interpreted as the internal signals of the cortical neurons carrying much more implicit information than just the evoked sensory response. At the systems level, the need for such implicit signals can be understood from the perspective of the natural behavioral repertoire that the cortical network was adapted to control. I.e., the animal may need to blink, lick, make eye movements unrelated to the stimuli[31], adjust body position and the relative muscle tones across muscles, etc, and may in addition continually plan ahead for when it may be an appropriate situation to execute these and other adjustments as well as preparing for other future behavioral choices. In fact, much of what we refer to as 'spontaneous' brain activity may rather reflect such internal processes, which is supported from observations on the broadcasting of thirst signals across the entire central nervous system, for example[32]. In the present study, the approach was to first map out the state space of the spontaneous data, in terms of the activity distributions across the recorded neuron population and explore to what extent different types of sensory perturbations pushed the system state either outside its regular high-dimensional state space or into parts of that state space that were visited less frequently spontaneously. Although we only tested tactile and visual inputs, we expect that all other modalities will also impact the state of the S1 cortical networks[13,33,34], just as tactile information appears to be detectable across all parts of the cortex outside S1[14,35].

Manipulating internal cortical states have been shown to alter the processing of given sensory inputs[16–18,36]. Further evidence that internal cortical processing is deeply involved in the structure of the recorded signal was provided by Nguyen et al.[37] (2023) who showed that any sensory input is repeatedly followed by internal replay, which gradually alters the cortical population responses to the given sensory inputs, i.e., an internally driven representational drift. Our observation that input-specific activity perturbations changed the state for a very long time after the termination of the input (Fig. 5) is in line with this. In fact, this is what would be expected given that the cortical and thalamocortical network is extensively globally interconnected[14,17–19,38,39]. In principle, if such a recursive network was deterministic, i.e., noise-free, the sensory input would impact the activity of the network system forever. But since the biological brain isn't noise-free[40,41], the impact of the sensory perturbations will eventually die out. The long duration of these poststimulus perturbations of the cortical state likely reflects the physiological infrastructure of thought processes (i.e., the internal temporal evolution of the brain activity in the absence of sensory input), which despite the anesthesia appeared to be supported on the timescale of at least a second. Making information about previous specific inputs resident within the time-evolving internal network activity is a potential substrate of a working memory. In fact, the definitions in the neuroscience literature map very well to our observations: 'Working memory broadly describes those short-term mnemonic processes that maintain task-relevant information for guiding the on-going or upcoming decisions…'[42]. Similarly, the pioneering work of Goldman-Rakic used as a partial definition of neurons contributing to a potential working memory function those neurons that remain active even when the information they code for is no longer present[43,44], i.e., very similar to what we observed here.

We used PCA to characterize, time-step-by-time-step, the dominant activity distributions across populations of up to 70 simultaneously recorded S1 cortical neurons as a measure of the dimensionality of the neuron population behavior. But how good a measure is PCA in this context? As previously pointed out, because of its inherent assumption of linearity in the brain network, 'PCA may overestimate the number of independent variables to describe [the neuron population data]'[1]. The argument is that since the brain networks are likely to operate nonlinearly, nonlinear dimensionality reduction methods (of which there are many different examples in recent literature, see above) may more accurately reflect the intrinsic dimensionality of the brain neuron population behavior. This argument, however, has two caveats. First, this statement relies on the assumption that linear dimensionality reduction methods (i.e., PCA) are compared with an optimal non-linear dimensionality reduction method. There is a lack of evidence suggesting any specific non-linear dimensionality reduction method is an optimal one, evident by the diversity of the non-linear dimensionality reduction methods that are employed within the field. The fact that the resultant dimensions are lower than for a linear method doesn't imply that the discarded dimensions were irrelevant for the brain. Secondly, as the rat cortex contains 25 million neurons, our recordings as well as any neuron activity recording to date (max 10,000's of neurons) primarily run the risk of hugely underestimating the complexity and dimensionality of the brain circuitry activity.

The following insightful comment, 'While an experimenter may apply linear decoders to find information about a hypothesized latent variable in a certain brain area, there is no a priori reason to assume that the brain relies on such decoders'[1] in fact applies to all of the non-linear dimensionality reduction methods as well. The only exception would be for the one true dimensionality reduction method which reflects how the biological brain works, which is yet to be discovered. Because nonlinear dimensionality reduction methods work (just as linear methods do; i.e., this study) it doesn't mean that the algorithm(s) used for the reduction in any way reflect how the brain circuitry operates. Just as Jazayeri & Ostojic (2021) states, 'without concrete computational hypotheses, it could be extremely challenging to interpret measures of dimensionality' –i.e., if we don't know the computational architecture of the brain circuitry it will be extremely challenging to understand its dimensionality, regardless of which embedding method we use. But there is no fundamental difference between linear and nonlinear reduction algorithms in this regard as they both represent arbitrary implicit assumptions of brain computational architectures used to derive approximations of data with high complexity. A nonlinear reduction method that effectively describes the data in a visualization may do so because it tends to magnify features of the data that are in line with a lower dimensional interpretation and discard features that are not in line – a linear method with higher dimensions could be less at risk of causing this problem.

A potential limitation of our study is that the main conclusion of high-dimensional behavior of S1 neuron populations was based on data from 5 experiments. However, those included a total of 274 neurons, each measured across several millions of time bins (as we used ms resolution), and there was no difference in any of the experiments regarding the main conclusion, i.e., that there was a high-dimensional spontaneous state space that collapsed when the temporal resolution of the neural recording data decreased.

Unlike many recent studies, which focused on finding the most efficient low-dimensional embeddings of their neuron population data as a means to visualize the behavior of the neural system, our data instead underscores the importance of considering the neural representations as high-dimensional. For example, when we do, we see that even a limited S1 neuron population can separate fine temporal nuances of visual input and that the population keeps longer term internal signals related to the specific sensory inputs, similar to a kind of working memory. Even though many higher order dimensions of the brain circuitry representations may make a difference only in more selective contexts and could thus easily be missed with excessive dimensionality reduction, the ability to handle multiple contexts is a hallmark of biological brain processing. The representations in higher dimensions are likely needed to confer the rapid and adaptable flexibility and learning that can be observed in any study of behavioral complexity.

## Methods

### Ethics statement

Ethical approval for this research was obtained in advance from the local animal ethics committee in Lund/Malmö (Ethical permit ID: M13193-2017). The study was conducted in accordance with the local legislation and institutional requirements.

### Surgical procedure

Adult male Sprague-Dawley rats ($N = 5$, 320–430 g) were initially sedated with isoflurane gas (3%) for 1–2 min. Anesthesia was then induced with a mixture of ketamine (40 mg/kg) and xylazine (4 mg/kg), administered intraperitoneally. Continuous anesthesia during the experiment was maintained through an intravenous catheter in the right femoral vein, delivering a mixture of Ringer acetate and glucose mixed with ketamine and xylazine (20:1 ratio) at a rate of approximately 5 mg/kg/h. For Neuropixels recordings, a craniotomy of $2 \times 2$ mm was performed over the somatosensory forepaw cortical area. Following the probe insertion, the brain area was covered with a 1% agarose solution to prevent dehydration. Throughout the procedure, the anesthesia level was closely monitored to ensure a light yet sufficient anesthetic depth (as confirmed by the regular absence of withdrawal reflexes upon pinching the hind paws). Additionally, the irregular occurrences of synchronized brain activity, a sign of deep sleep[45], were continuously observed to confirm the appropriate anesthesia level. The choice of anesthetic was based on its known property to preserve patterns of sequential neuron activity recruitment across diverse forms of UP states compared to the awake state[46]. Following the completion of the experiment, the animals were euthanized with an overdose of pentobarbital.

### Neuropixels recordings and selection of neurons

We recorded spiking activity across primary somatosensory (S1) cortex with Neuropixels 2.0 silicon probes. S1 recordings were made in the forepaw region with coordinates -1.0–1.0 mm relative to bregma. The tip of the probe (384 recording channels) was inserted but only the upper 180 channels were used for analysis as these were located within the cortex, which is approximately 1.8 mm deep in the rat. Neuropixels recordings were digitized (30 us sampling time per channel) and processed using the Kilosort2.5 Matlab package for spike sorting. Every unit detected was visually inspected in the Phy library, focusing on spike shape, frequency, and amplitude. Units with spike firing frequencies below 0.8 Hz or those confused with the stimulation artifacts were manually deselected. Units that had more than 1% of their inter-spike-intervals (ISI) within 2 ms were considered non-isolated due to refractory period violation. Units were split or deselected based on ISI plot analysis.

The level of responsiveness to tactile stimulation was evaluated for each neuron using peristimulus time histograms (PSTHs) of the evoked responses to repeated tactile stimulations (see below). For the PSTHs, spike responses were first binned at 2 ms. The baseline activity was obtained for each individual neuron by calculating the average spike frequency in the 300 ms prestimulus time window. If the spike frequency exceeded the baseline activity by two standard deviations (SDs) for two consecutive time bins in the 300 ms poststimulus time window, the neuron was considered as responsive to the tactile stimulation and was included in the analysis.

### Sensory stimulation

In many recent studies of neuron population activity, researchers have employed stimuli with more naturalistic properties, for example, different types of movies in visual processing studies. As we have previously described in detail for haptic stimuli[10,14], naturalistic stimuli in awake animals may suffer from shortcomings in terms of reproducibility. In brief, there is no way to control that the same conditions of muscle activation and dynamic mechanical skin state are reproduced across repetitions even if the stimuli from the point of view of the experimenter look the same. A similar problem exists for visual stimuli, where the muscles controlling gaze direction, focal plane, and pupil size in the awake state vary at a high pace and thereby is a source of substantial variation in retinal photoreceptor activation across

trials[47,48]. To minimize the variation of which sensors were activated across repetitions of the same stimulus, we here used tactile and visual stimuli designed to activate the sensors in the same spatiotemporal patterns across all repetitions[10,28]. This allowed a high precision analysis because any systematic response variation now solely would be due to factors intrinsic to the processing network rather than differences in sensor activation.

Tactile stimulation was delivered as spatiotemporal patterns of electrical skin activation through four pairs of intracutaneous needle electrodes (channel# 1–4). The four patterns (which were labeled F5, F10, F20, F ∞ ; see REF 10) mimic hypothetical tactile afferent activation patterns for four different types of mechanical skin-object interactions[10]. The needle electrodes were inserted into the volar side of the second digit of the forepaw with 2–3 mm distance between each pair, and 1 mm or less within the pair. The stimulation patterns were delivered as pulses with intensities of 0.5 mA and durations of 0.14 ms (DS3 Isolated Stimulator, Digitimer, UK). The patterns were delivered in pre-defined random order separated by about 1.8 s (random intervals). Pattern durations were between 200 ms and 340 ms. Each tactile input pattern was repeated 200 times in each experiment, hence a total of 800 tactile stimuli was provided in each experiment.

For visual stimulation, we used a custom apparatus with four adjacent 7000 mcd LEDs (red, blue, green, and yellow) to deliver four unique spatiotemporal patterns (vA, vB, vC, vD). The apparatus was positioned at a 30° angle from the midline of the animal (sagittal plane), 4–5 cm from the left eye. Stimulation patterns were multicolor pulse trains comprising sequences of colored light pulses delivered in a pre-defined random order. Each sequence included two pulses of each color, for a total of 8 pulses. Each pulse had 5 ms duration, 1 mW power, and a 25 ms interval until the onset of the next pulse. We used a set of 4 different visual stimulation patterns, each with a total duration of 215 ms. Visual stimulation patterns were repeated in randomized order with 1.8 s intervals. In the regular protocol of visual stimulation, each stimulation pattern was repeated 200 times, and there was hence a total of 800 visual stimuli in each experiment.

In addition to experiments ($N = 3$) alternating between tactile stimulation patterns and visual stimulation patterns, we also did experiments ($N = 3$) with visuo-tactile stimulation patterns. The visuo-tactile stimulation patterns were visual pulse trains consisting of simultaneous activation of all four colors (i.e., these were non-patterned visual stimuli) and tactile stimulation patterns. Here, the visual pulse train consisted of five multicolor pulses where all four LEDs were activated in each pulse. The pulses had 5 ms duration and a 50 ms interval until the onset of the next visual stimulation pulse. The visual pulse trains started simultaneously with the tactile stimulation patterns, which were varied across the four tactile patterns used in this study. The visuo-tactile stimulation patterns were repeated 200 times in each experiment. Since there were 4 tactile patterns, there were in each experiment a total of 800 visuo-tactile stimuli that could be compared with 800 tactile stimuli composed of the same 4 tactile patterns, presented in the same experiments.

### Statistics and reproducibility

Representation of neuronal spike data as time continuous functions. The main analysis was based on the activity distributions across the neuron population during spontaneous activity compared with evoked activity. Spontaneous activity was defined as all the time points of the 300 ms prestimulus time windows, whereas evoked activity was the 300 ms poststimulus time windows, except in the analysis of tactile patterns where it was extended to 1000 ms poststimulus onset.

In order to represent spike responses from individual neurons as continuous functions, spike trains in predefined analysis time windows were binned at 1 ms (or at 5, 10, 50, 100, 300 ms for the tests with down-sampled neural data). Then we averaged 50 sequential responses (pre+poststimulus time windows) in order to avoid potential classification errors caused by infrequently spiking neurons. The resulting responses were convolved with a 10 ms Gaussian kernel, which was motivated by that 10 ms is close to the apparent spike time resolution under similar experimental conditions (Oddo et al., 2017).

Principal component analysis. PCA was used to identify the covariance patterns in the activity distribution across the neuron population. Note that we have previously explored the separation of different spatiotemporal tactile input patterns of S1 neurons, and spatiotemporal visual input patterns of V1 neurons, using PCA applied to the full temporal profiles of the responses of each individual neuron recorded in sequence[10,14,17–19,28]. Here, instead, we looked at the momentary activity distribution across the simultaneously recorded population of neurons but did not analyze the full temporal profile of the responses. Hence, the present paper provides a different type of analysis than in those previous papers.

In the present study, the PCs were based on all 1 ms data points of the spontaneous activity, i.e., the 300 ms prestimulus baseline activity recorded across 800 or 1600 prestimulus time windows (i.e., depending on whether we had a total of 5 or 10 min of spontaneous activity, which varied depending on the experiment). Hence the state space of the spontaneous activity was defined from 800 or 1600 times 300 data points per experiment. PCs explaining up to 95% of the variance of the spontaneous data were extracted and used to compare spontaneous and evoked activities.

KNN analysis. To determine if evoked spike responses had a different activity distribution across the neuron population than the spontaneous data, the location of data points in the high-dimensional PC space was compared using k-Nearest Neighbour analysis (kNN). KNN uses the Euclidean distances between the data points and can hence identify the nearest neighbors for each individual data point. In a binary decision task, kNN can quantify to what extent the nearest neighbors of an evoked data point are predominantly data points of the same category (evoked data points versus spontaneous data points) and vice versa. If the number of data points for which this is true exceeds 50% (the chance level in a binary decision task), then it can be said that the distributions are different between the two conditions.

The k-nearest neighbor (kNN) analysis was performed with $k = 5$ nearest neighbors. Data points were split into training and test sets (50/50) and the kNN analysis was repeated 50 times, each with a new random training/test split. The average result was reported as the decoding accuracy (i.e., the values shown in the confusion matrices). Note that for the kNN analysis to work properly, we needed the same number of spontaneous data points as we had evoked data points. If there were only 800 stimulations of one condition, for example, 800 tactile stimulations, then we used 800 randomly selected time windows of spontaneous activity (out of 1600 available) for the comparison. To assess the significance of the obtained estimate of the decoding accuracy, we also conducted kNN analyses with shuffled data, where labels were randomly reassigned between evoked and spontaneous data points. This shuffling was repeated 50 times for each data set.

As a measure for decoding accuracy, we used the F1 score. First, precision and recall were calculated with True Positives (TP), False Positives (FP) and False Negatives (FN):

$$precision = \frac{TP}{TP + FP}$$

$$recall = \frac{TP}{TP + FN}$$

With the precision and recall parameter for each of the $8 \times 8$ matrices, the F1-scores were calculated:

$$F1 = 2 \times \frac{precision \times recall}{precision + recall}$$

For each experiment, the separation of the spontaneous activity from tactile and visually evoked activity, respectively, i.e., the F1 score, was calculated using different numbers of PCs, from 1 PC up to the number of PCs explaining 95%. If an experiment required 50 PCs to reach the 95% limit,

this resulted in 50 F1 scores for the tactile analysis and 50 F1 scores for the visual analysis from each population of S1 neurons. To visualize how the value of the F1 scores varied with the number of PCs, both sets of scores and the cumulative variance were plotted as logarithmic scales for an example experiment.

KNN analyses across specific time bins. We evaluated the activity from visual patterns using kNN on 10 ms time windows for data evoked across 300 ms in a series of 10 ms time windows poststimulus onset, resulting in 30 kNN analyses for each experiment. Each of the 30 kNN analyses for individual experiments was repeated 50 times, each time with a new random training/test split. The mean F1 score was calculated for each of the 30 time windows. To display the overall accuracy for individual experiments, the average F1 score ±2 standard deviations (SDs) of across all time windows, was displayed in a bar plot for individual experiments. As a control, all kNN analyses were also performed with shuffled labels resulting in a chance level F1 score. The mean F1 score ±2SDs across all time windows was also displayed in the bar plot as an average from all three experiments.

A similar approach as for the visual patterns was used to analyze activity evoked by the tactile patterns. Here, however, we analyzed the time spanning from 200 ms prestimulus to 1000 ms poststimulus (1200 ms) in a series of 10 ms time windows, resulting in 120 results. The mean F1 score ±2SDs of the 50 repetitions for each of the 120 kNN analyses was displayed for individual time windows separately for each experiment. The vector of F1 scores ±2SDs was smoothed with a rolling average of three bins. Each of the time window analyses was also performed with shuffled labels. To analyze if there was a statistically significant difference between the kNN decoding accuracy before and after stimulation onset, we compared the distribution of F1 scores in five separate two-sided paired sample t-tests. To do this, we grouped the F1 scores from all 50 repetitions from the 20 decoding analyses in the 200 ms prestimulus time window for all experiments together ($N = 3000$) and did the same for scores in five equally sized time windows poststimulation onset (1–200, 201–400, 401–600, 601–800, and 801–1000). The prestimulus F1 score distribution was then compared to the poststimulus distributions in five separate paired sample t-tests. Before each test, the distributions were checked for normality through visual inspection. All tests were followed by an estimate of effect size calculated with Cohen's $d$:

$$d = \frac{Mean_1 - Mean_2}{SD_{pooled}}$$

PCA+kNN for visuo-tactile of individual neurons. We have previously compared the responses of individual neurons to different sensory stimulation patterns using PCA on the temporal response profiles of individual responses (see ref. 28 for example). In the present paper, we repeated this analysis to provide a comparison with the population level covariance pattern analysis (PCA+kNN) we used in the present paper. To estimate decoding accuracy in individual neurons we extracted spike times for individual neurons in the 300 ms time window poststimulus onset to all four tactile stimulation patterns and all four visuo-tactile stimulation patterns. Spike trains were convolved with a 5 ms Gaussian kernel and split into a training and test set 50/50. PCs explaining 95% of the variance across stimulus repetitions were extracted from the training set only. Using kNN, we then obtained the F1 scores for each pair of visuotactile/tactile stimulation pattern. Reponses from one example neuron to the four tactile patterns and the four visuo-tactile patterns are displayed as a PSTH with a KDE in Supplementary Fig. 2.

CNN decoding. As an additional control, a convolutional neural network (CNN) approach was used as a control analysis. CNNs were trained to separate spontaneous activity from tactile and visual evoked activity, respectively. Here, the input data was provided in the same format as in the PCA, with k number of neurons and t number of binned time points which

was then split into 30% test and 70% training data. Labels of the data were one hot encoded to ensure that the distance between the two categorical variables (spontaneous versus tactile, spontaneous versus visual, respectively) was the same. A shallow convolutional neural network was constructed consisting of two convolutional layers followed by two dense layers. The first convolutional layer consisted of $1 \times 5 \times 5$ filters where the dimensions represented neuron x time x number of filters in given order. Each filter in this layer, convolved with a padding of 4 followed by a pooling layer with a window size of 3 and padding of 1, was fed into a ReLu activation function followed by a second convolution layer with the dimensions of $1 \times 5 \times 10$. Both convolutional layers had their weights initialized according to normalized Xavier weight initialization[49] where the weights were uniformly sampled in the range:

$$\frac{-\sqrt{6}}{m+n} + \frac{\sqrt{6}}{m+n}$$

where n was the number of inputs to the node and m was the number of outputs from the node. The dense layer, which consisted of a node number of 160 multiplied with the number of neurons, with a ReLu activation function, was followed by a second dense layer with two output layers and a softmax output function for categorical classification. Weight updates were performed based on a categorical cross entropy loss with a gradient descent update rule where the starting learning rate was 0.01 and was halved after 50 epochs until the learning rate was down to $10^{-5}$ and the weights with the lowest loss on the test dataset were saved. This whole training process was repeated 24 times to create an average confusion matrix from which the F1 score were calculated in the same way as described above.

## Reporting summary

Further information on research design is available in the Nature Portfolio Reporting Summary linked to this article.

## Data availability

The source data is freely available on the Figshare server, https://figshare.com/articles/dataset/Data_for_High-dimensional_cortical_signals_reveal_rich_bimodal_and_working_memory-like_representations_among_S1_neuron_populations_/26369989. Additional data are available on reasonable request to the corresponding author.

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

## Acknowledgements

This research was funded by the EU Horizon 2020 Research and Innovation Program under Grant agreement No. 861166 (INTUITIVE).

## Author contributions

S.S.K. and H.J. planned and designed the study. S.S.K. and H.J. designed the main analyses. S.S.K. conducted the experiments and the analysis. K.K. contributed to the main analysis and designed the CNN control analysis. S.S.K. and H.J. wrote the article. S.S.K., H.J., and K.K. authors contributed to the article and approved the submitted version.

## Funding

## Competing interests

The authors declare no competing interests
