## [Transparent Peer Review file · Communications Biology]

High-dimensional cortical signals reveal rich bimodal and working memory-like representations among S1 neuron populations

Corresponding Author: Ms Sofie Kristensen

Version 0:

Reviewer comments:

Reviewer #1

(Remarks to the Author)

In this study, the authors investigate the effects of combining sensory input from two modalities (tactile and visual) on the activity of the somatosensory cortex (S1). They use Neuropixels probes in S1 of an anesthetized rodent while they present the sensory input. They find that the tactile input is affected by the visual input. The experiments were carefully conducted, and the findings were potentially interesting. The figures and the writing are arcane, convoluted, and difficult to follow. There are many abbreviations, for instance, in the figures, that are not explained and make it difficult to follow. I suggest extensive revision of text and figures to abide by common principles of scientific writing and presentations.

Major concern:

Overall, several concepts are not or poorly defined, and the presented data are highly processed. Less processed data should have direct evidence of the claimed effects. All concepts should be clearly explained and referenced. I suggest writing concepts in the caption rather than abbreviations.

The term "Complexity" is not properly defined in the introduction. It is a scientific concept, not a research field.- and different fields have different definitions. The authors claim that "here we assessed the signal complexity...." But no metric of complexity is presented anywhere, although the authors seem to equate complexity with high-dimensionality in the signal. If this is the case, it should be clearly stated and preferably regarding previous literature using similar metrics. However I do not see any reason to use the term complexity in this context- it will likely cause more confusion for the general reader.

Minor:

Title: the link to working memory is relatively weak. I do not recommend including weak and speculative conclusions in the title.

Line 110 and elsewhere: Neuropixel -> Neuropixels

Fig. 1:A: the caption says it's a raster, but it is not a raster, it is the estimated firing rates.

Fig 2A: The F10, F20 and so on are not explained in the caption. What is that? Is it related to F1?

Fig. 2B: The grey dots on the grey background make it difficult to see the point. I suggest a different color. PC space of what? What does each dot represent? It's also not explained.

"Fig 2C: "F1 scores" -> "decoding accuracy (F1 scores)".

Reviewer #2

(Remarks to the Author)

General comments:

In this study, the authors investigate the representation of sensory information by neuronal population dynamics in the primary somatosensory cortex (S1) of anesthetized rats using high-density extracellular recordings. They apply PCA with high temporal resolution (1 ms) as a dimensionality reduction method to explore how sensory information is represented. The authors show, as expected, that the neuronal dynamics evoked by sensory stimuli differ clearly from the dynamics of spontaneous activity. Less expected is the discovery that a significant portion of the information about sensory input is contained in higher dimensions of neuronal activity. In particular, higher dimensions of neural activity in S1 can significantly decode visuo-tactile stimuli from tactile stimuli or even visual stimuli alone, indicating that multimodal integration occurs even in primary sensory areas and is represented in the higher dimensions of neuronal dynamics. Finally, the authors also

show that sensory information can be retained in population dynamics long after stimulus presentation, thus providing a possible substrate for short-term memory.

This article presents a very interesting analytical and conceptual approach that challenges many recent publications claiming low-dimensionality representation in the dynamics of neuronal populations in the brain. The final discussion is particularly interesting. However, the main claims of the article are partly limited by the fact that the experiments are carried out on anesthetized animals and by a small number of experiments (N=3). Additionally, further analysis could be done to strengthen the authors' point, in my opinion.

Specific comments:

1) The authors argue that performing experiments under anesthesia allows for better control and reproducibility of the sensory inputs applied to the animal. Which is correct. However, many studies have shown that most of the variability in the sensory evoked neuronal activity actually comes from fluctuations in the spontaneous activity which are actually higher under anesthesia. In fact, the state of wakefulness is often characterized by reduced variability in neuronal activity and more reliable sensory-evoked activity. Furthermore, one could argue that regardless of the variability in the sensory evoked activity, animals can detect sensory stimuli with high accuracy and reliability, indicating that the brain is in fact built to cope with this variability in neuronal activity. The neuronal population dynamics are certainly dramatically different in the anesthetized brain compared to the awake state and the authors should be more careful in their conclusions and discuss this limitation in the discussion. Ultimately, the interesting results reported in this article should be confirmed and validated in awake animals.

2) In general, the study suffers from very low N numbers, which makes it difficult to correctly assess the significance of certain effects. This is particularly the case for the lasting change in neuronal dynamics after the sensory stimulus. The authors could add new data or find ways to better estimate the statistical significance of the results.

3) One of the most interesting points of the article in my opinion - which is clearly addressed in the discussion - is the idea that the high dimensionality of neuronal population dynamics is often underestimated in many studies by the readout of neuronal activity at low temporal resolution (i.e. either due to the inherent limitation of calcium sensors for optical measurements, or due to the integration of the neuronal activity over long temporal windows). The authors rightly point out that this temporal filtering can significantly underestimate the information contained in the higher dimensions of population activity. The authors therefore chose to study population dynamics with a very high temporal resolution, at 1 ms. Although I completely agree with this logic, I would still say that a temporal resolution of 1 ms may be too short. If we consider a downstream neuron reading the activity of the recorded population of neurons, we can argue that due to synaptic integration, this neuron is likely capable of integrating synaptic inputs in the range of several tens of ms. This delay will be even longer if we consider a recurrent neural network. I think this consideration should be taken into account in the discussion.

4) Additionally, I would suggest adding more in-depth analysis to reinforce this point. It would be really interesting to perform some of the analyses with different temporal resolutions. For instance, I expect that as temporal resolution decreases (longer time bins), the dimensionality of population dynamics (i.e., the number of PCs containing significant information about the sensory stimulus) will also decrease.

5) A less important point, simply to make the data and analyses easier for the reader to understand, would be to show more neural activity, i.e. PSTHs for sensory-evoked responses: different tactile and visual stimuli, visuo-tactile or tactile alone (which could be shown in supplementary figures).

Minor points:

1) In the analysis of the separation of tactile patterns showing a persistence of sensory information long after the stimulus, a better estimate of the baseline and chance level would be important. In Figure 5C, it appears that the F1 score is already higher than the Shuffle version before the onset of the stimulus, which could indicate some sort of long-term bias or drift in the activity of the neuronal population. Figure 5C should show a longer baseline. Additionally, there should be a Shuffle line for each experiment (i.e. 3 Shuffle lines). Finally, it would be important to better define the times at which the F1 score is significantly higher than chance level (or the baseline!). Similarly, Figure 4C should also show a baseline value (just before stimulus onset).

2) I found it particularly interesting that some information about the nature of the sensory stimulus appears to be contained in the higher order PCs. I think this would require further clarification and exploration. For example, in Figure 2, the authors show that visuo-tactile and tactile stimuli can be distinguished by PCs #10-12 without further justification. Similarly, in Figure 4, PCs 10 to 12 are used to separate the touch patterns, but then in Figure 5 A-B, the first 3 PCs are shown and finally in Figure 5C, all 50 PCs are used. It would be really interesting to more systematically quantify which PCs contain relevant information for the different sensory inputs, somewhat along the lines of Figure 1.

3) Were the neurons included in the PCA selected? Only responsive neurons, as indicated line 139-140? If so to what extent could that impact the estimation of the dimensionality of the neuronal population dynamic? Some neurons may not respond to the stimulus but significantly contribute to the dimensionality of the spontaneous activity.

4) Was the PCA performed on the raw firing rates or was some normalization applied. The legend of Figure 1A indicates some normalization, however the scale bar of panel A seems to indicate that the firing rate (Hz) is displayed?

5) In figure 5B it is not really clear what each dot represents? Is it one time bin for a single trial?

IN SUMMARY:

Some additional experiments to increase the N numbers would help to convince but would not be mandatory.

Add to discussion about some experimental limitations.

Add further analyses and quantifications using different temporal resolutions and better quantifying the dimensionality of population dynamics (i.e. how many/which PCs contain significant information).

Show more neural activity.

Sylvain Crochet

Author Rebuttal letter:

Reviewer #1 (Remarks to the Author):

In this study, the authors investigate the effects of combining sensory input from two modalities (tactile and visual) on the activity of the somatosensory cortex (S1). They use Neuropixels probes in S1 of an anesthetized rodent while they present the sensory input. They find that the tactile input is affected by the visual input. The experiments were carefully conducted, and the findings were potentially interesting. The figures and the writing are arcane, convoluted, and difficult to follow. There are many abbreviations, for instance, in the figures, that are not explained and make it difficult to follow. I suggest extensive revision of text and figures to abide by common principles of scientific writing and presentations.

We are grateful for the many comments that helped us make important clarifications in our paper. We have revised the figures and the writing to make them less convoluted and easier to follow, and explained abbreviations in places where they were missing.

Major concern:

Overall, several concepts are not or poorly defined, and the presented data are highly processed. Less processed data should have direct evidence of the claimed effects. All concepts should be clearly explained and referenced. I suggest writing concepts in the caption rather than abbreviations.

We have addressed the concern of poorly explained concepts and tried to explain and reference them better, including to spell them out in the caption rather than using abbreviations. We have added examples of less processed data, Supplementary Figure 2, in addition to what was previously shown in Figure 1A. As we point out there, it can be noted that comparisons between the different peristimulus histograms of individual neuron responses are suggestive of some potential differences, but these differences were not strong enough to be statistically significant. However, such differences became more systematic when we instead focused our analysis to the activity distribution patterns across the S1 neuron populations. We now point out in several places that these results would not have been detected without the population level analysis that we made. The data processing is the basis for us being able to report something fundamentally new to the literature, and is the most direct way to describe the claimed effects. But we have tried to explain this more clearly.

The term "Complexity" is not properly defined in the introduction. It is a scientific concept, not a research field.- and different fields have different definitions. The authors claim that "here we assessed the signal complexity...." But no metric of complexity is presented anywhere, although the authors seem to equate complexity with high-dimensionality in the signal. If this is the case, it should be clearly stated and preferably regarding previous literature using similar metrics. However I do not see any reason to use the term complexity in this context- it will likely cause more confusion for the general reader.

Yes, that is a good point. We have better defined the concept complexity and explained the link between high-dimensional effects in the data and this concept. Both are important to convey the findings of this paper, i.e. complexity is what high-dimensional systems can give rise to, and as indicated by papers that we cited, complexity is widely present in neural data and behavior if it is carefully looked for (see also new added reference in the discussion by Musall et al 2019 'Single-trial neural dynamics are dominated by richly varied movements'). Basically, the many advantages of complexity are the key reason why it is important to investigate the dimensionality of the population-level neural signal, and a key message of our paper is that that dimensionality has been greatly underestimated in the literature.

As we now elaborate in the Intro, complexity is an area of science where there are specific definitions of what it means. There are several journals dedicated to the study of complex systems ('complexity') and for example PLoS has just launched a new journal within this rapidly growing research field (i.e. 'New Complexity Science Journal from PLoS' (PLoS Complex Systems), starting 2024. The call for papers includes network theory, and the field is thereby highly relevant for neuroscience). The Wikipedia page on complexity states that 'A complex system is a system composed of many components which may interact with each other... Complex systems are systems whose behavior is intrinsically difficult to model due to the dependencies, competitions, relationships, or other types of interactions between their parts or between a given system and its environment. Systems

that are "complex" have distinct properties that arise from these relationships, such as nonlinearity, emergence, spontaneous order, adaptation, and feedback loops, among others. Because such systems appear in a wide variety of fields, the commonalities among them have become the topic of their independent area of research. In many cases, it is useful to represent such a system as a network where the nodes represent the components and links to their interactions.'

Hence, in the intro we have added:

'In general, a complex system is composed of many components, which can interact with each other and thereby create dependencies and other types of relationships that cause distinct system-level properties to arise, one of them being that the system output has high complexity. A complex system can be described as a network where the nodes represent the components and the links between them represent their interactions. In fact, the brain seemingly fulfills many of the definitions of a complex system, where the neurons would be the nodes and the synaptic connections between them create dependencies. This would be true if it can be shown that the activity of the individual neurons have unique properties from the other neurons – if so, that would also mean that the activity distribution patterns between the neurons would be high-dimensional.'

Also, we note that the other reviewer stresses the importance of these elements: 'This article presents a very interesting analytical and conceptual approach that challenges many recent publications claiming low-dimensionality representation in the dynamics of neuronal populations in the brain. The final discussion is particularly interesting.'

Minor:

Title: the link to working memory is relatively weak. I do not recommend including weak and speculative conclusions in the title.

We have expanded the section on working memory, to show that our results fulfills the main criteria that has been used in the literature (starting with Goldman-Rakic in 80's, but also very recent papers) to label neural activity as being contributing to working memory. In fact, across all scientific fields (psychology, computer science, neuroscience) a definition of working memory is information that is present in the system and can be used to impact current decision-making. In our Figure 5 (which has now been improved based on comments by the other reviewer), we clearly show that the tactile information is present across the population of neurons, and as such would impact the interpretation of the next tactile event.

Line 110 and elsewhere: Neuropixel -> Neuropixels
Corrected

Fig. 1:A: the caption says it's a raster, but it is not a raster, It is the estimated firing rates.
Corrected

Fig 2A: The F10, F20 and so on are not explained in the caption. What is that? Is it related to F1?
Clearly a bad mistake by us, thanks for pointing it out. We now explain it in the Methods, and in the caption

Fig. 2B: The grey dots on the grey background make it difficult to see the point. I suggest a different color. PC space of what? What does each dot represent? It's also not explained.
Clarified, the dots have been made darker

"Fig 2C: "F1 scores" -> "decoding accuracy (F1 scores)".

Thanks, Corrected

Reviewer #2 (Remarks to the Author):

General comments:

In this study, the authors investigate the representation of sensory information by neuronal population dynamics in the primary somatosensory cortex (S1) of anesthetized rats using high-density extracellular recordings. They apply PCA with high temporal resolution (1 ms) as a dimensionality reduction method to explore how sensory information is represented. The authors show, as expected, that the neuronal dynamics evoked by sensory stimuli differ clearly from the dynamics of spontaneous activity. Less expected is the discovery that a significant portion of the information about sensory input is contained in higher dimensions of neuronal activity. In particular, higher dimensions of neural activity in S1 can significantly decode visuo-tactile stimuli from tactile stimuli or even visual stimuli alone, indicating that multimodal integration occurs even in primary sensory areas and is represented in the higher dimensions of neuronal dynamics. Finally, the authors also show that sensory information can be retained in population dynamics long after stimulus presentation, thus providing a possible substrate for short-term memory.

This article presents a very interesting analytical and conceptual approach that challenges many recent publications claiming low-dimensionality representation in the dynamics of neuronal populations in the brain. The final discussion is particularly interesting. However, the main claims of the article are partly limited by the fact that the experiments are carried out on anesthetized animals and by a small number of experiments (N=3). Additionally, further analysis could be done to strengthen the authors' point, in my opinion.

We thank the reviewer for the very insightful comments and that there is appreciation of the conceptual advances this paper provides.

Specific comments:

1) The authors argue that performing experiments under anesthesia allows for better control and reproducibility of the sensory inputs applied to the animal. Which is correct. However, many studies have shown that most of the variability in the sensory evoked neuronal activity actually comes from fluctuations in the spontaneous activity which are actually higher under anesthesia. In fact, the state of wakefulness is often characterized by reduced variability in neuronal activity and more reliable sensory-evoked activity. Furthermore, one could argue that regardless of the variability in the sensory evoked activity, animals can detect sensory stimuli with high accuracy and reliability, indicating that the brain is in fact built to cope with this variability in neuronal activity. The neuronal population dynamics are certainly dramatically different in the anesthetized brain compared to the awake state and the authors should be more careful in their conclusions and discuss this limitation in the discussion. Ultimately, the interesting results reported in this article should be confirmed and validated in awake animals.

We agree that the variability in the sensory evoked response for single neurons could be higher in anesthetized animals. This would be an expected consequence of the more pervasive presence of large amplitude slow-wave activity, which can at least episodically impact the neuron activity to a great degree, and thereby potentially making the apparent input-driven response variability at the single neuron level more prominent (although in the awake animal there may be time-varying ongoing internal brain processing unrelated to the stimulus that could also induce high apparent response variability). But for a population level analysis like we do here, where the covariance patterns of activity distributions are measured, it is likely that it works the other way around. That is, a population of neurons would tend to be more correlated with each other, because the thalamic inputs they share tend to become more synchronized (i.e. this is the reason we can measure slow oscillatory EEG waves in deep sleep and anesthesia). More correlated neurons, as in the anesthetized state, means less diverse covariance patterns. Therefore, as the latter quantity is what we measure, our data on the dimensionality of the neuron population behavior is almost certainly an underestimate relative to how this system would behave in the awake state. There is in fact some experimental evidence for this in non-anesthetized animals (de Oliveira et al (2022) 'Off-manifold coding in visual cortex revealed by sleep'; <https://www.biorxiv.org/content/10.1101/2022.06.10.495710v2>). In this paper, it is stated that '...these results suggest that awake activity is composed of the low-dimensional dynamics observed in SWS [slow-wave sleep, related to the state under our light anesthesia] with off-manifold activity added. We found similar results in multiple other brain regions, suggesting this may be a general feature of neural population activity.' Hence, in neuron populations across various brain regions, the awake state is characterized by higher dimensionality than the slow wave sleep state, as we argue above. We clarified our discussion on this.

2) In general, the study suffers from very low N numbers, which makes it difficult to correctly assess the significance of certain effects. This is particularly the case for the lasting change in neuronal dynamics after the sensory stimulus. The authors could add new data or find ways to better estimate the statistical significance of the results.

The main conclusion, high-dimensional behavior of neuron populations, is based on data from 5 experiments with a total of 274 neurons each measured across several millions of time bins (as we use ms resolution). There was no difference in any of the animals regarding the main conclusion, i.e. that there was a high-dimensional spontaneous state space that collapsed when the temporal resolution of the neural recording data is decreased (see below, we thank the reviewer for this excellent suggestion). Also there was no difference across any of the neuron populations in the other measured quantities either, which we now have made clearer by increasing the amount of displayed data in the figures 4 and 5. We also introduced a better estimate of the statistical significance of the results in Figs 4 and 5, to make it clear that these data are robust.

It is common in papers where 100's of neurons are recorded in parallel use very few experiments. For example, this paper which we cited, and which is also using Neuropixels recordings, had only N=3 experiments: <https://www.nature.com/articles/s41586-021-04268-7> and there are many more examples in the literature of this. But in agreement with the reviewer's final remarks, we added a short limitations section on this issue.

3) One of the most interesting points of the article in my opinion - which is clearly addressed in the discussion - is the idea that the high dimensionality of neuronal population dynamics is often underestimated in many studies by the readout of neuronal activity at low temporal resolution (i.e. either due to the inherent limitation of calcium sensors for optical measurements, or due to the integration of the neuronal activity over long temporal windows). The authors rightly point out that this temporal filtering can significantly underestimate the information contained in the higher dimensions of population activity. The authors therefore chose to study population dynamics with a very high temporal resolution, at 1 ms. Although I completely agree with this logic, I would still say that a temporal resolution of 1 ms may be too short. If we consider a downstream neuron reading the activity of the recorded population of neurons, we can argue that due to synaptic integration, this neuron is likely capable of integrating synaptic inputs in the range of several tens of ms. This delay will be even longer if we consider a recurrent neural network. I think this consideration should be taken into account in the discussion.

We agree with the general notion that it is certainly not guaranteed that any neuronal network would always be able to make use of information on such a fast time base as 1 ms (but we think there is no difference between recurrent connections and feed-forward connections in this regard, as the signal of the recurrent connection has the effect that it has already been processed 'baked in' into its signal, and the timing of that processed signal is what matters rather than the absolute delay from the sensor). In fact, considering the extremely high dimensionality we found in a mere 70-neuron recording makes it hard to imagine what dimensionality could have been found across all the 25 mln neurons across the neocortex of the rat. With the new Fig 1F that we added, thanks to the reviewer's suggestion, we think it is now clear that the brain will lose dimensionality the lower the temporal resolution of its signal processing. Other than that, we feel that we cannot make any definite statements regarding the temporal resolution of signal processing within the cortical network. We added the following statement though:
'And, as illustrated in Figure 1F, that dimensionality would also depend on the actual temporal resolution of the information representation within the cortical networks, but that issue would require a very different investigation to be settled.'

4) Additionally, I would suggest adding more in-depth analysis to reinforce this point. It would be really interesting to perform some of the analyses with different temporal resolutions. For instance, I expect that as temporal resolution decreases (longer time bins), the dimensionality of population dynamics (i.e., the number of PCs containing significant information about the sensory stimulus) will also decrease. Exactly, that's a great point. We added the analysis of how our measures depend on the temporal resolution of the data. We added a new panel F in Figure 1 to show this. We also show in an added analysis in Figure 2D that the information about the presence of visual input totally vanishes when the temporal resolution goes from 1 ms to 300 ms. Those new panels really underscore our point, and we thank the reviewer for this good suggestion.

5) A less important point, simply to make the data and analyses easier for the reader to understand, would be to show more neural activity, i.e. PSTHs for sensory-evoked responses: different tactile and visual stimuli, visuo-tactile or tactile alone (which could be shown in supplementary figures).

OK, we have added a supplementary figure 2 showing the tactile alone and then the visuo-tactile. As can be seen from these PSTHs, the effect of adding the visual input is subtle, it is there, but at the single neuron level it is not statistically significant. That significance only appears when we combine the information across the neuron population, as we do in the main figure.

Minor points:

1) In the analysis of the separation of tactile patterns showing a persistence of sensory information long after the stimulus, a better estimate of the baseline and chance level would be important. In Figure 5C, it appears that the F1 score is already higher than the Shuffle version before the onset of the stimulus, which could indicate some sort of long-term bias or drift in the activity of the neuronal population. Figure 5C should show a longer baseline. Additionally, there should be a Shuffle line for each experiment (i.e. 3 Shuffle lines). Finally, it would be important to better define the times at which the F1 score is significantly higher than chance level (or the baseline!). Similarly, Figure 4C should also show a baseline value (just before stimulus onset).

OK, good points, we have added all of this.

2) I found it particularly interesting that some information about the nature of the sensory stimulus appears to be contained in the higher order PCs. I think this would require further clarification and exploration. For example, in Figure 2, the authors show that visuo-tactile and tactile stimuli can be distinguished by PCs #10-12 without further justification. Similarly, in Figure 4, PCs 10 to 12 are used to separate the touch patterns, but then in Figure 5 A-B, the first 3 PCs are shown and finally in Figure 5C, all 50 PCs are used. It would be really interesting to more systematically quantify which PCs contain relevant information for the different sensory inputs, somewhat along the lines of Figure 1.

The reason that we show some of the higher order PCs is exactly the point that the reviewer raises, this is extremely interesting and likely very surprising. The particular choice of PCs for illustration is actually random. But then all the systematic analysis is done using the F1 score measure, which takes all the PCs into account. But we cannot illustrate all the high order PCs, so illustrations by necessity have to be samples. Note that we already showed in Fig. 1E, for the two different types of inputs, exactly how much each consecutive PC added – this figure shows that there is no reason to suspect that any particular one of the higher order PCs makes a step-wise difference. It is more of an accumulative improvement the more of high order PCs are included in the quantification. And this is also our main argument for why considering the high dimensions of the data is important to not discard. We now point this out more clearly in relevant parts of the Results and captions.

3) Were the neurons included in the PCA selected? Only responsive neurons, as indicated line 139-140? If so to what extent could that impact the estimation of the dimensionality of the neuronal population dynamic? Some neurons may not respond to the stimulus but significantly contribute to the dimensionality of the spontaneous activity.

Yes, this is true, but this choice has more practical reasons. A neuron that rarely fire is very likely to further increase the dimensionality of the system, as it creates covariance patterns that otherwise does not occur when it actually fires. If anything, therefore, this choice of ours actually leads to an underestimate of the dimensionality of the data.

4) Was the PCA performed on the raw firing rates or was some normalization applied. The legend of Figure 1A indicates some normalization, however the scale bare of panel A seems to indicate that the firing rate (Hz) is displayed?

PCA only cares about the shape of the covariance pattern, i.e. the activity distribution across the neuron population. Hence, no normalization is applied. The top panel of Figure 1A displays raw firing data where normalization makes it easier to read the figure.

5) In figure 5B it is not really clear what each dot represents? Is it one time bin for a single trial?

Clarified in the caption

IN SUMMARY:

Some additional experiments to increase the N numbers would help to convince but would not be mandatory. Add to discussion about some experimental limitations.

We added to discussion about some experimental limitations (related to the N numbers) in the concluding remarks.

Add further analyses and quantifications using different temporal resolutions and better quantifying the dimensionality of population dynamics (i.e. how many/which PCs contain significant information). Show more neural activity.

We did both of these additions, and the temporal resolution figure panel very much improved our paper.

Sylvain Crochet

We thank the reviewer again for very good inputs

Version 1:

Reviewer comments:

Reviewer #1

(Remarks to the Author)

The authors have made an effort to address my concerns. Regarding the concept “complexity”, the authors seem to still struggle to get a proper handle on exactly what it means, how to measure it and make reference to the literature. Again they write e.g. line 56 “...the system output has high complexity.” Without explaining how to assess high versus low complexity. In the same sentence the authors also define complexity with the term complexity in the same sentence, thus making a circular definition.

They write “A complex system can be described as a network where the nodes represent the components and the links between them represent their interactions”. Does that mean all networks are complex systems? And systems which are not networks, cannot be complex? Of course not, but they leave the reader guessing.

Regarding keeping in the concept, the authors argue that complexity is an important concept and therefore should be kept in the manuscript. I do agree that the concept of complexity is important but failing to defining it properly or make reference to the prior work on the topic, only confuses the reader. If the authors cannot get to the essence of how a “complex” system is different from a non-complex system and how to assess this, I advise taking this topic out of the manuscript to avoid confusion.

Generally I still find the writing needs more clarity- if the PI has not already been involved in the stringency of writing I highly recommend doing so.

Reviewer #2

(Remarks to the Author)

The authors have made appropriate revisions to their article. I have no further comments. The article is now acceptable for publication in Communications Biology.

Sylvain Crochet

Author Rebuttal letter:

We are grateful for the helpful comments of the reviewers

We addressed the remaining concern of Reviewer #1 by clarifying the link between complexity and high dimensionality in neuronal representations in the Introduction. The definitions closely adhere to those on the Wikipedia page that defines the properties of complex systems.
